# Autoencoder-based Initialization for Recurrent Neural Networks with a Linear Memory

## Abstract

Orthogonal recurrent neural networks address the vanishing gradient problem by parameterizing the recurrent connections using an orthogonal matrix. This class of models is particularly effective to solve tasks that require the memorization of long sequences. We propose an alternative solution based on explicit memorization using linear autoencoders for sequences. We show how a recently proposed recurrent architecture, the Linear Memory Network, composed of a nonlinear feedforward layer and a separate linear recurrence, can be used to solve hard memorization tasks. We propose an initialization schema that sets the weights of a recurrent architecture to approximate a linear autoencoder of the input sequences, which can be found with a closed-form solution. The initialization schema can be easily adapted to any recurrent architecture. We argue that this approach is superior to a random orthogonal initialization due to the autoencoder, which allows the memorization of long sequences even before training. The empirical analysis shows that our approach achieves competitive results against alternative orthogonal models, and the LSTM, on sequential MNIST, permuted MNIST and TIMIT.

## 1 Introduction

Several sequential problems require the memorization of long sequences of patterns. As an example, a generative model for music should be able to memorize long sequences of notes and be able to repeat them, as it is typically done in musical pieces. RNNs and LSTMs struggle to solve even simple memorization tasks (Arjovsky et al., 2015; Graves et al., 2014). Therefore, it is important to study alternative solutions to this problem.

Orthogonal RNNs are a class of recurrent architectures that solve the vanishing gradient problem by constraining the recurrent connections to be an orthogonal or unitary matrix (Arjovsky et al., 2015). They are particularly effective to solve long memorization tasks Henaff et al. (2016).

In this paper, we address the problem of memorization with orthogonal RNNs and linear autoencoders. Our objective is to *find a solution to the problem of memorization of long sequences*. The memorization of long sequences with orthogonal models can require a large number of hidden units, increasing the hidden state size and the cost in time and space.

If we assume that the input sequences in the training data lay in a low-dimensional manifold, as is typically believed for real-world data, then we can train an autoencoder with a small number of hidden units, sufficient to encode the entire sequence.

If we restrict ourselves to the linear case, we can compute the optimal autoencoder with a closed-form solution (Sperduti, 2013). This result can be exploited to initialize recurrent architectures to approximate the linear autoencoder for the input sequences. In our experiments we use the RNN (Elman, 1990) and the Linear Memory Network (LMN) (Bacciu et al., 2019). The LMN with the autoencoder initialization is a recurrent architecture equivalent to the Elman RNN but able to solve the vanishing gradient problem and memorize input sequences with a minimal hidden state.

We test our approach on classic benchmarks for orthogonal RNNs, showing that our proposed approach behaves similarly to orthogonal architectures on pure memorization tasks, and even improving the performance on real-world datasets. Finally, we show that the model can also be used in situations where a strict orthogonal parameterization struggles (Vorontsov et al., 2017), like the TIMIT benchmark (Garofolo et al., 1993).

Work on orthogonal models typically focuses on the properties of an orthogonal parameterization at the backward step, to address the vanishing gradient. In our work instead, we focus on the forward step, by investigating the effect of an orthogonal parameterization against an autoencoder-based solution. For these reasons, we are not particularly interested in enforcing exact orthogonality constraints, but in the study of the effectiveness of an autoencoder-based memorization mechanism. Our proposed approach requires the memorization of the entire sequence within the hidden state activations. It is necessary to note that this approach can be quite inefficient whenever the underlying task does not require complete memorization of the input sequences. In this situation, the hidden state size necessary to encode the entire sequence could be much larger than the minimal hidden state size necessary to solve the problem. Therefore, it is fundamental to allow the model to diverge from the orthogonality or the autoencoder solution by only imposing soft orthogonality constraints. Nonetheless, we believe that even when complete full memorization is not necessary, the autoencoder initialization can help to speed up training convergence by allowing the model to remember long sequences, which can then be gradually forgotten during training if deemed uninformative.

In summary, the main contributions of the paper are:

- the proposal of a novel initialization schema designed for the explicit memorization of long sequences;
- highlighting the connection between orthogonal models and linear autoencoders for sequences;
- an empirical analysis that shows the effectiveness of our proposed initialization schema to solve tasks that require long memorization.

## 2 LINEAR MEMORY NETWORKS

The Linear Memory Network (LMN) (Bacciu et al., 2019) is a recurrent neural network that computes a hidden state $\boldsymbol{h}^t$ and a separate memory state $\boldsymbol{m}^t$. The hidden state $\boldsymbol{h}^t$ is computed with a fully connected layer from the input and the previous state followed by an activation function. The memory state $\boldsymbol{m}^t$ is computed with a linear transformation of the current hidden state and the previous memory state. The equations for the model update are the following:

$$
\begin{array}{rcl}
\boldsymbol{h}^t & = & \sigma(\boldsymbol{W}_{xh}\boldsymbol{x}^t + \boldsymbol{W}_{mh}\boldsymbol{m}^{t-1}) \\
\boldsymbol{m}^t & = & \boldsymbol{W}_{hm}\boldsymbol{h}^t + \boldsymbol{W}_{mm}\boldsymbol{m}^{t-1}
\end{array}
$$

where $\boldsymbol{W}_{xh}, \boldsymbol{W}_{mh}, \boldsymbol{W}_{hm}, \boldsymbol{W}_{mm}$ are the model's parameters and $\sigma$ is a non-linear activation function ($tanh$ in our experiments), and $\boldsymbol{m}^t$ is the final output of the layer. The linear recurrence can be used to control the memorization properties of the model, as we will show in the subsequent sections.

The linearity of the memory update can be exploited to guarantee a constant propagation of the gradient by constraining $W_{mm}$ to be an orthogonal matrix. This property allows the LMN to avoid the vanishing gradient problem. Notice that to guarantee the constant gradient propagation it is also necessary to truncate the gradient between $m^{t-1}$ and $h^t$ by setting $\frac{\partial h^t}{\partial m^{t-1}} = 0$. This approach is similar to the truncated backpropagation proposed originally for the LSTM (Hochreiter & Schmidhuber, 1997), which is necessary to guarantee that an LSTM without a forget gate propagates the gradients without vanishing effects. The modification to the training algorithm is trivial to implement with the current automatic differentiation packages, as done in (Kanuparthi et al., 2018). However, in this work, we focus on the effect of the orthogonality on the forward propagation and therefore we only use the full gradient in our experimental results.

### 2.1 EQUIVALENCE LMN-RNN

The linearity of the memory update of the LMN might seem a major limitation of the architecture. However, it is easy to show that an RNN can always be rewritten as an equivalent LMN. Given an RNN with parameters $\boldsymbol{V}$ and $\boldsymbol{U}$ such that $\boldsymbol{h}_{rnn}^t = \sigma(\boldsymbol{V}\boldsymbol{x}^t + \boldsymbol{U}\boldsymbol{h}_{rnn}^{t-1})$, we can define an equivalent

LMN such that $\boldsymbol{m}^t = \boldsymbol{h}_{rnn}^t \forall t$ by setting the parameters in the following way: $\boldsymbol{W}_{xh} = \boldsymbol{V}, \boldsymbol{W}_{mh} = \boldsymbol{U}, \boldsymbol{W}_{hm} = \boldsymbol{I}, \boldsymbol{W}_{mm} = 0$.

Therefore, the linearity does not limit the expressivity of the LMN. Differences between the two architectures however become important during training. The linear update of the LMN without proper regularization can be affected by the exploding gradient. This problem is less frequent in RNNs due to the non-linearity (which often favors the vanishing gradient). In preliminary experiments, we found that training LMNs without any kind of regularization can lead to instability during the training. Fortunately, the gradient propagation can be controlled in the linear recurrence by tuning the spectral radius $\rho$ of the recurrent matrix $W_{mm}$, for example, by using an orthogonal parameterization. In our experiments we use soft orthogonality constraints by adding a regularization term $\lambda\|\boldsymbol{W}^\top \boldsymbol{W} - I\|^2$ as in Vorontsov et al. (2017). Alternatively, it is possible to build a fading memory system by imposing $\rho < 1$ (Tino et al., 2004).

Another possible cause of instability is the uncontrolled growth of the norm of the memory state $m^t$. The LSTM solves this problem by using a forget gate (Cummins et al., 2000) to reset the memory cell activations. This is not an issue in RNNs with a bounded nonlinear activation. We overcome this problem by adding to the loss function a regularization term $\beta\frac{1}{T}\sum_{t=1}^{T}(\|\boldsymbol{h}_t\|_2 - \|\boldsymbol{h}_{t-1}\|_2)^2$ that penalizes the norm of the hidden activations as in Krueger & Memisevic (2015).

## 3 INITIALIZATION WITH A LINEAR AUTOENCODER FOR SEQUENCES

The linear recurrence of the LMN allows to exploit results from the field of linear autoencoder for sequences (LAES) (Sperduti, 2013; 2015; Pasa & Sperduti, 2014).

In this section, we show how to initialize the LMN and RNN models using a linear autoencoder for sequences trained on the input samples. This initialization allows the resulting model to encode the input sequences into a minimal hidden state.

### 3.1 LINEAR AUTOENCODER FOR SEQUENCES

A linear autoencoder for sequences (LAES) (Sperduti, 2013) is a linear model composed of a linear autoencoder, which is a linear dynamical system, and a linear decoder. The autoencoder takes as input a sequence $\boldsymbol{x}^1, \ldots, \boldsymbol{x}^T$ and computes an internal state $\boldsymbol{m}^t$ with the following equations:

$$\boldsymbol{m}^t = \boldsymbol{A}\boldsymbol{x}^t + \boldsymbol{B}\boldsymbol{m}^{t-1}$$

$$\begin{bmatrix} \tilde{\boldsymbol{x}}_t \\ \tilde{\boldsymbol{m}}_{t-1} \end{bmatrix} = \boldsymbol{C}\boldsymbol{m}_t,$$

where $\boldsymbol{A}$ and $\boldsymbol{B}$ are the encoder parameters, $\boldsymbol{C}$ is the decoding matrix and $\tilde{\boldsymbol{x}}^t$ and $\tilde{\boldsymbol{m}}^{t-1}$ are the reconstructions of the current input and the previous memory state.

Sperduti (2013) provides a closed-form solution for the optimal linear autoencoder. The corresponding decoder matrix $\boldsymbol{C}$ can be reconstructed from the autoencoder parameters as $\boldsymbol{C} = \begin{bmatrix} \boldsymbol{A}^\top \\ \boldsymbol{B}^\top \end{bmatrix}$.

### 3.2 LAES INITIALIZATION

By means of the linear autoencoder for sequences, we can construct a simple linear recurrent model which uses the autoencoder to encode the input sequences within a single vector, the memory state of the autoencoder. To predict the desired target, we can train a linear layer that takes as input the states of the autoencoder to predict the target. The equations for the model are as follows:

$$\boldsymbol{m}^t = \boldsymbol{A}\boldsymbol{x}^t + \boldsymbol{B}\boldsymbol{m}^{t-1} \tag{1}$$

$$\boldsymbol{y}^t = \boldsymbol{W}_o \boldsymbol{m}^t, \tag{2}$$

where $\boldsymbol{A}$ and $\boldsymbol{B}$ are the parameters of the autoencoder trained to reconstruct the input sequences and $\boldsymbol{W}_o$ are the parameters of the readout trained to the predict the target. Figure 1a shows a schematic view of the architecture.

Pasa & Sperduti (2014) propose to initialize an RNN with the linear RNN defined in Eq. 1 and 2:

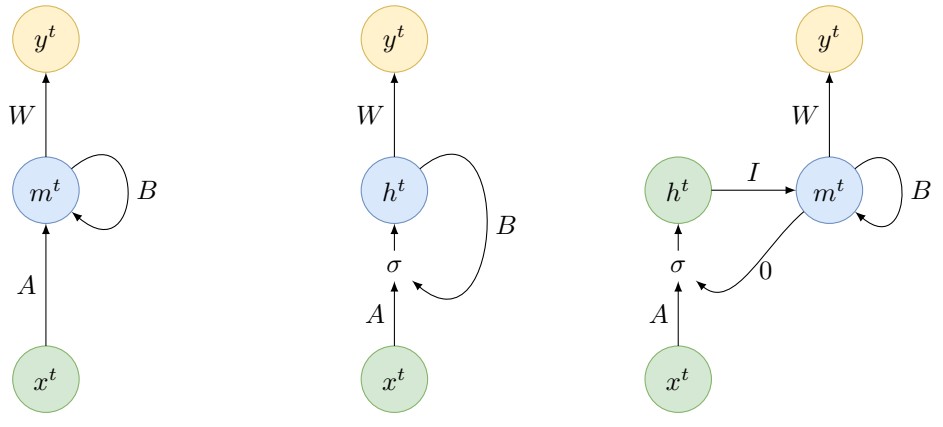

(a) Linear RNN.  (b) RNN with LAES initialization.  (c) LMN with LAES initialization.

Figure 1: Linear RNN and initialization schemes for RNN and LMN. Notice how the nonlinearity only affects the input in the LMN with LAES initialization. Blue nodes highlight the LAES memory state.

$$\boldsymbol{h}^t = tanh(\boldsymbol{A}\boldsymbol{x}^t, +\boldsymbol{B}\boldsymbol{h}^{t-1}) \tag{3}$$

$$\boldsymbol{y}^t = \boldsymbol{W}_o\boldsymbol{h}^t \tag{4}$$

This initialization approximates the linear RNN. The quality of the approximation depends on the values of the hidden activations $\boldsymbol{m}^t$. With small values close to zero, the $tanh$ activation function is approximately linear, and therefore we obtain a good approximation of the autoencoder. However, for large values the correspondence between the autoencoder activations and the initialized RNN degrades quickly since the $tanh$ activation function enters into the nonlinear part and cannot represent values $> 1$ or $< -1$. In practice, we find that the correspondence between the linear autoencoder and the hidden states of the initialized RNN with a $tanh$ activation function tends to degrade quickly due to the accumulation of the error through time.

To solve this problem, we propose to adopt a similar initialization scheme to initialize the weights of an LMN using the linear RNN defined in Eq. 1 and 2. The LMN model is initialized as follows:

$$\boldsymbol{h}^t = \sigma(\boldsymbol{A}\boldsymbol{x}^t + \boldsymbol{0}\boldsymbol{m}^{t-1})$$
$$\boldsymbol{m}^t = \mathbf{I}\boldsymbol{h}^t + \boldsymbol{B}\boldsymbol{m}^{t-1}$$
$$\boldsymbol{y}^t = \boldsymbol{W}_o\boldsymbol{m}^t$$

where the parameters of the LMN are initialized to approximate the linear RNN: $\boldsymbol{W}_{xh} = \boldsymbol{A}, \boldsymbol{W}_{mh} = 0, \boldsymbol{W}_{hm} = \boldsymbol{I}, \boldsymbol{W}_{mm} = \boldsymbol{B}$. The initialization is only an approximation of the linear RNN but since the nonlinearity only affects the input transformation $\boldsymbol{A}\boldsymbol{x}^t$ and not the recurrent part $\boldsymbol{B}\boldsymbol{y}^t$, the approximation is closer to the original linear RNN. Figures 1b and 1c show a schematic view of the architectures. Experimental results in Section 5 provide a quantitative comparison between the two approaches. These results show that the accuracy of the initialized LMN is the same as the linear RNN.

## 4 DECODING WITH THE UNROLLED MODEL

To give an intuition on the operations enabled by the use of an autoencoder-based memorization mechanism, we discuss how it is possible to use the decoder to build an explicit model that is able to reconstruct the entire sequence, and to process the unrolled sequence with a feedforward layer that combines weights for each specific time delay. Let us assume to have the optimal autoencoder with parameters $\boldsymbol{A}$ and $\boldsymbol{B}$ trained on the input sequence $\boldsymbol{x}^1, ..., \boldsymbol{x}^T$. We further assume that the autoencoder has enough hidden units to have exactly zero reconstruction error. The output of the

entire model can be computed with a linear layer as follows:

$$y^t = W^o m^t, \tag{5}$$

where $y^t$ is the output vector at time $t$ and $W^o$ the parameters of the output layer. Given the hidden state $m^t$ at time $t$ of the autoencoder, we can reconstruct the input at time $t - k$ using the decoder $C = \begin{bmatrix} A^\top \\ B^\top \end{bmatrix}$. In fact, after $k$ applications of the decoder we find:

$$x^{t-k} = A^\top (B^\top)^{k-1} m^t, \tag{6}$$

which shows how to reconstruct any input vector $x^\tau$, $\tau = 1, \ldots, t-1$, from the autoencoder memory $m^t$. The output layer can reconstruct the entire input sequence using the decoder. This can be made explicit with a decomposition of $W^o$ as:

$$W^o = \begin{bmatrix} W^0 & ... & W^{t-1} \end{bmatrix} \begin{bmatrix} A^\top \\ ... \\ A^\top (B^\top)^{t-1} \end{bmatrix}$$

where $W^i$ is the matrix applied to the input at delay $i$. Using this decomposition, the output $y^t$ can be expressed as:

$$y^t = \begin{bmatrix} W^0 & ... & W^{t-1} \end{bmatrix} \begin{bmatrix} x^t \\ ... \\ x^1 \end{bmatrix}.$$

This resulting model requires explicit parameters for each time delay, with a total number of parameters that grows linearly with the length $T$ of the sequence (or the maximum dependency length $k < T$), and it is expensive to train. However, using the autoencoder, these two operations are combined within a single matrix multiplication.

Notice that in general $W^o$ will not be decomposed with this factorization. However, any model of this form can be potentialy learned by gradient descent (given a suitable loss function). Therefore it is in principle possible for an autoencoder-based model to reconstruct the entire sequence and process each element separately.

The unrolled model shows the expressiveness of the autoencoder. By comparison, an orthogonal parameterization contains the activations of the entire sequence but it is not possible, in principle, to separate them and reconstruct the original input. Furthermore, since the encoding is not explicitly optimized, memorization with an orthogonal parameterization might become inefficient and be prone to require a larger hidden state size.

## 5 EXPERIMENTS

In this section, we evaluate the performance of the orthogonal LMN and the proposed initialization scheme on synthetic tasks and real-world datasets: the copy task (Arjovsky et al., 2015), digit classification with sequential and permuted MNIST (Le et al., 2015), and framewise classification with TIMIT (Garofolo et al., 1993). These datasets are standard benchmarks for the assessment of orthogonal recurrent neural networks, and they offer the opportunity to evaluate the proposed approach in different settings. While for permuted MNIST orthogonal models reach state-of-the-art results, Vorontsov et al. (2017) and Jose et al. (2018) showed that orthogonality constraints can reduce the performance of the trained model on TIMIT. We also compare the LAES initialization for the RNN and the LMN, showing that the LMN is a better approximation in practical scenarios.

Each model is trained with Adam (Kingma & Ba, 2014), with a learning rate in $\{10^{-3}, 10^{-4}, 10^{-5}\}$ chosen by selecting the best model on a separate validation set. A soft orthogonality constraint $\lambda \| W^\top W - I \|^2$ is added to the cost function as in Vorontsov et al. (2017), with $\lambda$ chosen $\{0, 10^{-5}, 10^{-4}, 10^{-3}, 10^{-2}\}$.

### 5.1 COPY TASK

The copy task is a synthetic benchmark often used to test the ability of a model to capture long-term dependencies. We use the same setup as in Jose et al. (2018). The objective is the memorization

of a sequence of $S$ elements, which must be repeated sequentially after $T$ timesteps. The input is a sequence of $2S + T$ timesteps where the first $S$ elements are randomly generated from a set of $K$ elements, followed by $S - 1$ blank elements, an output delimiter, and $S$ blank elements. The target sequence contains $S + T$ blank elements followed by the first $S$ elements of the input sequence. To solve this task, the network must be able to memorize $S$ elements into its hidden state and remember them for $T$ timesteps. Each model is a single layer recurrent architectures with 100 hidden units and has been trained with 10 000 batches containing 64 samples each.

The copy task can be easily solved with linear orthogonal models. Henaff et al. (2016) showed a handcrafted solution for the problem. However, adding a nonlinearity makes the task difficult to solve even for orthogonal models (Henaff et al., 2016). The LMN solves this limitation by separating the nonlinearity used to compute the hidden activations from the linear memory update. As shown in Table 1, the LMN can solve the task even using a saturating nonlinearity ($tanh$ in our experiments). In more general terms, the LMN is able to combine nonlinear activations with the memorization properties of orthogonal linear models.

Table 1 shows the results on the copy task for $T = 100$ and $T = 500$, with $S = 10$, and $K = 8$. We compare the results to a memoryless baseline that outputs the blank element for $S + T$ timesteps and random elements for the last $S$ elements. The LMN and the linear RNN solve the task perfectly in both cases. The RNN with a $tanh$ nonlinearity fails to beat the baseline even with $T = 100$. The LSTM beats the baseline in both settings but does not solve the task optimally.

These results confirm our expectations that we can use the LMN architecture with a saturating nonlinearity even for tasks that require learning long-term dependencies.

Table 1: Test accuracy on the Copy task.

|                        | T=100 | T=500 |
| ---------------------- | ----- | ----- |
| RNN ($tanh$)           | FAIL  | -     |
| RNN (linear orthogonal)| 100   | 100   |
| LMN                    | 100   | 100   |
| LSTM                   | 99.7  | 99.4  |

## 5.2 SEQUENTIAL AND PERMUTED MNIST

Sequential and permuted MNIST (Le et al., 2015) are two standard benchmarks to test the ability of recurrent neural networks to learn long-term dependencies. They consist of sequences extracted from the MNIST dataset (LeCun, 1998) by scanning each image one pixel at a time, in order (sequential MNIST) or with a random fixed permutation (permuted MNIST). We use these datasets to compare the LAES initialization to a random orthogonal initialization. A validation set is extracted by sampling 10 000 random images from the training set and separating them from the training data to use them to perform the model selection.

These datasets provide an ideal setting to test a pure memorization approach like the LAES initialization since the output layer can use the final memory state $m^t$ to reconstruct the entire sequence of hidden activations.

Table 2 shows the results on the two MNNIST benchmarks. We compare the RNN and LMN with the LAES initialization and a random orthogonal initialization with several orthogonal models. We also show the results of the linear RNN used to initialize the models with the LAES initialization. All the models have 128 hidden units and a single hidden layer. Each model has been trained for 100 epochs with a batch size of 64.

The best results are achieved by the AntisymmetricRNN (Chang et al., 2019), while the LMN with the LAES initialization obtains slightly lower results, but still improves compared to the results of the other orthogonal models.

Table 2: Test accuracy on sequential MNIST and permuted MNIST.

| | $n_h$ | Sequential MNIST Valid. | Test | Permuted MNIST Valid. | Test |
|---|---|---|---|---|---|
| FC uRNN(Wisdom et al., 2016) | 512 | 97.5 | 96.9 | 94.7 | 94.1 |
| EURNN(Jing et al., 2016) | 1024 | - | - | 94.0 | 93.7 |
| KRU(Jose et al., 2018) | 512 | 96.6 | 96.4 | 94.7 | 94.5 |
| LSTM(Jose et al., 2018) | 128 | 98.1 | 97.8 | 91.7 | 91.3 |
| ASRNN(Chang et al., 2019) | 128 | - | 98.0 | - | **95.8** |
| ASRNN(gated)(Chang et al., 2019) | 128 | - | **98.8** | - | 93.1 |
| Linear RNN | 128 | 87.1 | 86.6 | 84.6 | 84.2 |
| RNN (ortho init.) | 128 | 79.2 | 77.2 | 85.0 | 84.9 |
| RNN (LAES init.) | 128 | 97.3 | 97.0 | 92.4 | 91.8 |
| LMN (ortho init.) | 128 | 92.0 | 92.6 | 89.1 | 88.7 |
| LMN (LAES init.) | 128 | 98.7 | **98.5** | 96.3 | **95.4** |

The strength of the LAES initialization is the result of the ability of the LAES to efficiently compress the entire sequence and the high performance that can be obtained on MNIST using a linear model[1] and the linear RNN.

## 5.3 TIMIT FRAMEWISE CLASSIFICATION

TIMIT is a speech recognition corpus (Garofolo et al., 1993). In our experiments we follow the setup of Jose et al. (2018) for the framewise classification problem (Graves & Schmidhuber, 2005). Differently from MNIST, this dataset is more difficult for orthogonal models than for non-orthogonal models. Vorontsov et al. (2017); Jose et al. (2018) have shown that orthogonality constraints can be detrimental on this dataset. Therefore, it is ideal to study the performance of the LMN and the LAES initialization in a more challenging setup. All the models have been trained for 50 epochs with a batch size of 1. We compare models with a single layer and a varying number of hidden units to keep the number of parameters approximately constant across the models (about 400k).

Table 3 shows the results of the experiments.

Table 3: Test accuracy on TIMIT. Results for the RNN, LSTM and KRU-LSTM are taken from Jose et al. (2018)

| | $n_h$ | Valid. | Test |
|---|---|---|---|
| RNN(Jose et al., 2018) | 600 | 65.84 | 64.53 |
| LSTM(Jose et al., 2018) | 300 | 65.99 | 64.56 |
| KRU-LSTM(Jose et al., 2018) | 2048 | 66.54 | 64.81 |
| LMN-ortho | 420 | 67.4 | **66.1** |
| LMN-laes | 420 | 65.0 | 63.4 |

We found that without any regularization the norm of the memory state $m^t$ tended to grow indefinitely, which degraded the final performance. We did not find the same problem on the MNIST dataset. A penalization on the memory state norm Krueger & Memisevic (2015) was sufficient to solve this issue. We selected $\lambda$ by cross-validation in $\{0, 1, 10, 100\}$. It is important to note that, while on this benchmark the best result is achieved by the LMN, the best model does not use the LAES initialization. The best configuration for the LMN use an orthogonal initialization but does not use soft orthogonality constraints since the best configuration is obtained when the hyperparameter for the soft orthogonality constraints is $\lambda = 0$. This confirms the other results in the literature and shows that the architecture is general enough to model problems that do not require explicit memorization.

---

[1]results for a linear classifier can be found at `http://yann.lecun.com/exdb/mnist/`

### 5.4 LAES INITIALIZATION

In Section 3 we remarked that LAES initialization can incur in large errors due to the nonlinearity that affects the hidden state of the autoencoder. To verify if these problems happen in practical scenarios, we trained a linear autoencoder with 128 hidden units on sequences from sequential and permuted MNIST and initialized an LMN and an RNN with it. Table 4 shows the results.

Table 4: Comparison between the LAES initialization on sequential and permuted MNIST for the RNN and the LMN using the test accuracy. The LMN is a better approximation and therefore reaches the same performance, while the RNN has a much lower accuracy.

|  | Sequential MNIST | | Permuted MNIST | |
| --- | --- | --- | --- | --- |
|  | TRAIN | VALID | TRAIN | VALID |
| linear-RNN | 85.5 | 87.2 | 82.9 | 84.6 |
| RNN (after init.) | 11.7 | 11.3 | 27.3 | 27.5 |
| LMN (after init.) | 85.5 | 87.2 | 83.0 | 84.7 |

The results show that the error of the RNN initialization is large enough to cause a significant drop in the accuracy, from 85.5 of the linear RNN to 11.7 of the initialized model. Conversely, the initialized LMN reaches the same accuracy of the linear RNN. Therefore, it seems that the linear recurrence is the model is necessary to achieve a good approximation of the autoencoder. The results on TIMIT show a similar trend.

## 6 RELATED WORK

Orthogonal RNNs solve the vanishing gradient problem by parameterizing the recurrent connections with an orthogonal or unitary matrix Arjovsky et al. (2015). Some orthogonal models exploit a specific parameterization or factorization of the matrix (Mhammedi et al., 2017; Jose et al., 2018; Jing et al., 2016) to guarantee the orthogonality. Other approaches constrain the parameters with soft or hard orthogonality constraints (Vorontsov et al., 2017; Zhang et al., 2018; Wisdom et al., 2016; Lezcano-Casado & Martínez-Rubio, 2019). Vorontsov et al. (2017) have shown that hard orthogonality constraints can hinder training speed and final performance.

Linear autoencoders for sequences can be trained optimally with a closed-form solution (Sperduti, 2013). They have been used to pretrain RNNs (Pasa & Sperduti, 2014). The LMN (Bacciu et al., 2019) is a recurrent neural network with a separate linear connection.

The memorization properties of untrained models are studied in the field of echo state echo networks (Jaeger & Haas, 2004), a recurrent model with untrained recurrent parameters. Tino et al. (2004) showed that untrained RNNs with a random weight initialization have a Markovian bias, which results in the clustering of input with similar suffixes in the hidden state space. Tiňo & Rodan (2013) and White et al. (2004) study the short-term memory properties of linear and orthogonal memories.

## 7 CONCLUSION

In this work, we studied the problem of building an autoencoder-based memorization mechanism. This system has the ability to encode and decode the input sequences to compute the desired target. Using results for the linear autoencoder for sequences, we showed how to initialize a recurrent neural network by approximating a linear autoencoder of the input sequences. The architecture exploits a linear recurrence to obtain a better approximation of the autoencoder. The results show that an autoencoder-based initialization can be effective for learning memorization tasks. In the future, we plan to extend this work by studying the effect of the autoencoder during training, possibly by enforcing the encoding of the input sequence even during the learning phase. Another possible avenue of research is the study of better optimization algorithms for the parameters of the linear component, where the linearity could be exploited to speed up the training process through dedicated learning algorithms.

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
