# OpenReview forum: "Autoencoder-based Initialization for Recurrent Neural Networks with a Linear Memory"
_ICLR.cc/2020/Conference — Reject_

### Official Review · AnonReviewer1 · 2019-10-09
**Official Blind Review #1**

**Rating:** 3

**Review:**

Summary:

The paper proposes an autoencoder-based initialization for RNNs with linear memory. The proposed initialization is aimed at helping to maintain longer-term memory and instability during training such as  exploding gradients (due to linearity).

Pros:

1. The paper is well written, the motivation and methods are clearly described.

Cons.

1. The authors claimed the proposed method could help with exploding gradient  in training the linear memories. It would be helpful to include some experiments indicating that this was the case (for the baseline) and that this method does indeed help with this problem.

2. The experiments on the copy task only showed results for length upto 500, which almost all baseline models are able to solve. I am not too sure how the proposed initialization helps in this case.

3. TIMNIT is a relatively small speech recognition dataset. The task/ dataset does not require long-term memorization. It is nice to see that the initialization helps in this case. However, it is still a little how this experiment corresponds to the messsage that the authors are attempting to deliver at the end of the introduction.

4. In general, it seems that the experiments could be more carefully designed to reflect the contributions of the proposed method. Some suggestions for future edits are, more analysis on gradients, maybe more experiments on the stability of training such as gradients could help.

Minor:

1. There are some confusions, on P2 "we can construct a simple linear recurrent model which uses the autoencoder to encode the input sequences within a single vector", I think the authors meant encode the input sequences into a sequence of vectors? Equation 1 and 2 suggest that there is a vector m^t per timestep (as oppose to having 1 for the entire sequence).

2. Although the copy task was used in ((Arjovsky et al., 2015), I believe the original task was proposed in the following paper and hence this citation should properly be the correct one to cite here,

Hochreiter, Sepp and Schmidhuber, Jürgen. Long short-term memory. Neural computation, 9(8):
1735–1780, 1997.



**Experience Assessment:**

I have published one or two papers in this area.

**Review Assessment: Checking Correctness Of Derivations And Theory:**

I carefully checked the derivations and theory.

**Review Assessment: Checking Correctness Of Experiments:**

I carefully checked the experiments.

**Review Assessment: Thoroughness In Paper Reading:**

I read the paper at least twice and used my best judgement in assessing the paper.

---

> ### Author Response · Authors · 2019-11-14
> **Response to Reviewer #1**
>
>
> >>> 1. The authors claimed the proposed method could help with exploding gradient  in training the linear memories. It would be helpful to include some experiments indicating that this was the case (for the baseline) and that this method does indeed help with this problem.
> >>> 4. In general, it seems that the experiments could be more carefully designed to reflect the contributions of the proposed method. Some suggestions for future edits are, more analysis on gradients, maybe more experiments on the stability of training such as gradients could help.
>
> We agree that this are interesting experiments. We believe that it is especially useful to study the effect on the gradient and training stability when combined with the truncated backpropagation (e.g. as done in the LSTM paper). Unfortunately, we still do not have the final results on these experiments.
>
>
> >>> 2. The experiments on the copy task only showed results for length upto 500, which almost all baseline models are able to solve. I am not too sure how the proposed initialization helps in this case.
>
> We used the experiments on the copy tasks to show that the LMN architecture learns the copy task with a saturating nonlinearity (tanh). As far as we know, this is the only architecture that can do it, while most of the other models use variations of ReLUs.
>
>
> >>> 1. There are some confusions, on P2 "we can construct a simple linear recurrent model which uses the autoencoder to encode the input sequences within a single vector", I think the authors meant encode the input sequences into a sequence of vectors? Equation 1 and 2 suggest that there is a vector m^t per timestep (as oppose to having 1 for the entire sequence).
>
> The state vector of the LAES m^t can be used to reconstruct the entire input sequence x^1, … x^t. Therefore, each vector m^t encodes the entire subsequence x^1, …, x^t. We will update the paper to make this point clearer.
>
>
> >>> 2. Although the copy task was used in ((Arjovsky et al., 2015), I believe the original task was proposed in the following paper and hence this citation should properly be the correct one to cite here
>
> Thank you for noticing this, we will add the reference.

---

### Official Review · AnonReviewer2 · 2019-10-21
**Official Blind Review #2**

**Rating:** 1

**Review:**

Summary:
This paper proposes a new initialization method for recurrent neural networks. They first obtain the weight from a linear optimal autoencoder. And then they use the weight to initialize the Lieanr Memory Networks(LMN). Basically, this paper is a combination of [1] and [2].

Strength:
The method of initializing LMN using a linear RNN is natural and simple. (section 3.2)
The proposed initialization outperforms the baselines on the MNIST dataset.

Weakness:
What do you mean by the "optimal autoencoder"?
The performance on TIMIT is worse than the baseline methods.
The scale of the experiments is too small. Do you have any experiment results on any large dataset? e.g. Penn Treebank.


Reference:
[1] Pre-training of Recurrent Neural Networks via Linear Autoencoders
[2] Linear Memory Networks

**Experience Assessment:**

I have read many papers in this area.

**Review Assessment: Checking Correctness Of Derivations And Theory:**

I carefully checked the derivations and theory.

**Review Assessment: Checking Correctness Of Experiments:**

I assessed the sensibility of the experiments.

**Review Assessment: Thoroughness In Paper Reading:**

I read the paper thoroughly.

---

> ### Author Response · Authors · 2019-11-14
> **Response to Reviewer #2**
>
>
> >>> Weakness: What do you mean by the "optimal autoencoder"?
>
> We use a linear autoencoder because we can find the optimal solution (in the sense that it optimizes the mean squared error) with a closed-form solution. We approximate this solution by taking a fixed number of components.
>
>
> >>> The performance on TIMIT is worse than the baseline methods. The scale of the experiments is too small. Do you have any experiment results on any large dataset? e.g. Penn Treebank.
>
> We did not perform experiments on PTB, but our expectation is that models based on autoencoding are not a good choice for language modeling tasks. This can be seen by looking at the performance of orthogonal models on language modeling tasks [1,2], which are always inferior to gated models. Our guess is that language modeling does not require the memorization of long sequences, and it is probably sufficient to remember a small amount of information.
>
> [1]  Eugene Vorontsov, Chiheb Trabelsi, Samuel Kadoury, and Chris Pal. On orthogonality and learning
> recurrent networks with long term dependencies. In ICML, pp. 3570–3578, 1 2017. URL
> [2] Cijo Jose, Moustpaha Cisse, and Francois Fleuret. Kronecker Recurrent Units. In ICML, pp.
> 2385–2394, 5 2018. URL http://arxiv.org/abs/1705.10142.

---

### Official Review · AnonReviewer3 · 2019-10-25
**Official Blind Review #3**

**Rating:** 3

**Review:**

This paper proposes an initialization scheme for the recently introduced linear memory network (LMN) (Bacciu et al., 2019) and the authors claim that this initialization scheme can help improving the model performance of long-term sequential learning problems.

My concerns lie with the novelty of the proposed model and the insufficiency of the experiments. First, the LMN seems to be a simpler version of LSTM and it has no significant advantages compared with other recurrent structures introduced in the past several years.
Second, the autoencoder-based init scheme (Pasa&Sperduti, 2014) is not new while the only technical contribution of this paper is a minor change of this scheme so that it works for the LMN. In my opinion, combining these two (LMN and init scheme) can hardly be considered as a solid novelty contribution.
For the experiment part, the first two tasks are a bit toyish in 2019 and I have not seen any significant improvement gained from the proposed model. Even for the TIMIT dataset, the results are a bit far from state-of-the-art which makes the paper's claim less convincing.

Overall I think the novelty contribution is marginal and I suggest the authors to test their models on larger-scale real problems.

The writing is clear and easy to follow.

**Experience Assessment:**

I have published one or two papers in this area.

**Review Assessment: Checking Correctness Of Derivations And Theory:**

I carefully checked the derivations and theory.

**Review Assessment: Checking Correctness Of Experiments:**

I carefully checked the experiments.

**Review Assessment: Thoroughness In Paper Reading:**

I read the paper thoroughly.

---

> ### Author Response · Authors · 2019-11-14
> **Response to Reviewer #3**
>
> You can, of course, obtain the LMN from the LSTM equations by eliminating all the gates and making the CEC update through a generic linear function (in place of the sum). But these are not minor changes and radically change the inductive bias of the models.  The LSTM uses the forget gate to reset the cell state activations, while the LMN uses a more efficient encoding and does not forget past activations. The result of this choice is that the LMN is better on tasks that require the memorization of long sequences (e.g. copy task, MNIST). On different tasks, like language modeling, other architectures (like the LSTM) are probably a better choice since the problem does not require explicit memorization.
>
>
> You are correct that the initialization scheme come from a previous contribution.  However, what we show here is that such initialization is more coherent with the assumption and nature of the LMN, rather than for a RNN with non-linear memory. We also show how that the autoencoder initialization can help in some tasks (improving both the results of orthogonal and gated models) and hinder the performance in others. These results show that the proposed approach is complementary to LSTMs and other gated architectures, and that the two approaches can be used to solve different tasks of different nature (in terms of long and short-term memorization abilities). We are not aware of other researches investigating the differences between encoding/orthogonal-based approaches and gated models and the capabilities of the different memorization schemes (which, to the extent of our understanding, appears a relevant topic for ICLR).
>
>
> >>> For the experiment part, the first two tasks are a bit toyish in 2019
> We agree that some of the benchmarks are simple tasks. However, the chosen datasets are used to compare against classic benchmarks in the literature of orthogonal RNNs.  Most of the papers in the literature use pixel-MNIST.
> [1] copy task, pixel-MNIST, PTB
> [2] copy task, TIMIT, pixel-MNIST
> [3] pixel-MNIST, pixel-CIFAR10
> [4] copy, MNIST, TIMIT, PTB, MIDI
>
> If the reviewer is aware of different benchmarks allowing to compare with the related literature we will gladly consider it.
>
>
> >>> Even for the TIMIT dataset, the results are a bit far from state-of-the-art which makes the paper's claim less convincing.
> Thank you for noticing this. The results may seem poor because we do not use bidirectional models. This is done to compare against [4]. We will update the paper to highlight this fundamental architectural difference with the literature.
>
>
> [1]  Eugene Vorontsov, Chiheb Trabelsi, Samuel Kadoury, and Chris Pal. On orthogonality and learning
> recurrent networks with long term dependencies. In ICML, pp. 3570–3578, 1 2017. URL http://arxiv.org/abs/1702.00071.
> [2] Scott Wisdom, Thomas Powers, John R. Hershey, Jonathan Le Roux, and Les Atlas. Full-Capacity
> Unitary Recurrent Neural Networks. In NIPS, pp. 4880–4888, 10 2016. URL http://arxiv.org/abs/1611.00035.
> [3] Bo Chang, Minmin Chen, Eldad Haber, and Ed H. Chi. AntisymmetricRNN: A Dynamical System
> View on Recurrent Neural Networks. 2 2019. URL http://arxiv.org/abs/1902.09689
> [4] Cijo Jose, Moustpaha Cisse, and Francois Fleuret. Kronecker Recurrent Units. In ICML, pp.
> 2385–2394, 5 2018. URL http://arxiv.org/abs/1705.10142.

---

### Decision · Program_Chairs · 2019-12-19

**Decision:**

Reject

**Comment:**

The paper explores an initialization scheme for the recently introduced linear memory network (LMN) (Bacciu et al., 2019) that is better than random initialization and the approach is tested on various MNIST and TIMIT data sets with positive results.

Reviewer 3 raised concerns about the breadth of experiments and novelty. Reviewer 2 recognized that the model performs well on its MNIST baselines and had concerns about applicability to larger settings. Reviewer 1 acknowledges a very well written paper, but again raises concerns about the thoroughness of the experiments. The authors responded to all three reviewers, responding that the tasks were chosen to match existing work and that the approach is complementary to LSTMs to solve different tasks. Overall the reviewers did not re-adjust their ratings.

There remains questions on scalability and generality, which makes the paper not yet ready for acceptance. We hope that the reviews support the authors further research.